# Can Mammography and Magnetic Resonance Imaging Predict the Preoperative Size and Nuclear Grade of Pure Ductal Carcinoma In Situ?

**DOI:** 10.3390/diagnostics15141801

**Published:** 2025-07-17

**Authors:** Hülya Çetin Tunçez, Merve Gürsoy Bulut, Zehra Hilal Adıbelli, Ahmet Bozer, Bülent Ahmet Kart, Demet Kocatepe Çavdar

**Affiliations:** 1Department of Radiology, Izmir City Hospital, Izmir 35540, Turkey; adibellizehra@gmail.com (Z.H.A.); drahmetbozer@gmail.com (A.B.); bulentahmetkart@gmail.com (B.A.K.); 2Department of Radiology, Ataturk Training and Research Hospital, Izmir Katip Celebi University, Izmir 35360, Turkey; gursoymerve@yahoo.com; 3Department of Radiology, Izmir Faculty of Medicine, University of Health Sciences, Izmir 35540, Turkey; 4Department of Pathology, Izmir City Hospital, Izmir 35540, Turkey; demetcavdar@gmail.com

**Keywords:** ductal carcinoma in situ, mammography, magnetic resonance imaging, breast cancer

## Abstract

**Background/Objectives**: Thirty to fifty percent of ductal carcinoma in situ (DCIS) cases are high-grade and at risk of progressing to invasive carcinoma. The most important treatment-related risk factor for recurrence is the presence of residual DCIS. The aim of our study was to evaluate the relationship between size and imaging features on preoperative mammography and magnetic resonance imaging (MRI) and histopathological size and nuclear grade in patients with pure DCIS. **Methods**: Between 2015 and 2023, 90 patients who underwent surgery for DCIS, had no microinvasive/invasive component, and underwent a preoperative mammography and MRI were included in this study. **Results**: DCIS was detected in 91.1% of patients using mammography and 95.5% using MRI. Microcalcifications (MCs) were most common in mammography (85.4%). Thin pleomorphic and thin linear branching MCs were detected in 42% of high-grade DCIS, while amorphous (42%) MCs were most common in low-grade DCIS. In low-grade DCIS cases, a grouped distribution of MCs was observed most commonly (69%). There was a statistically significant difference between DCIS groups in terms of MC morphology and distribution (*p* = 0.043, *p* = 0.005, respectively). Diffusion restriction on MRI was associated with high-grade DCIS (*p* = 0.043). The tumor size was greater than the pathological size and correlated poorly with mammography and moderately with MRI. **Conclusions**: Compared to mammography, MRI is more effective in detecting and estimating the size of DCIS. Both methods overestimate tumor size compared to histopathological size. The nuclear grade is associated with a poor prognosis and local recurrence in DCIS.

## 1. Introduction

Ductal carcinoma in situ (DCIS) is a non-invasive form of carcinoma in which malignant cells are confined to a duct with an intact basement membrane [1]. As breast cancer screening with mammography has increased, the incidence of DCIS has also increased over time. With improved early detection capabilities, more women are being diagnosed with this precancerous lesion before it develops into invasive cancer. This increase has created new challenges for both diagnosticians and treatment planning teams in the management and clinical approach to DCIS. Pure DCIS accounts for 15–25% of all breast cancers [2]. Of these cases, 30–50% are high-grade DCIS and carry a high risk of progression to invasive carcinoma [3]. Therefore, the early diagnosis and appropriate management of high-grade DCIS is crucial to prevent disease progression and avoid unnecessary invasive treatment and costs. In patients with DCIS diagnosed at an early stage, breast-conserving surgery with appropriate surgical margins can be planned, which can positively affect the patient’s quality of life [4].

Currently, the standard treatment for DCIS is surgery and adjuvant treatment, including radiotherapy or endocrine therapy [5]. However, recent studies have shown that not all cases of DCIS develop into invasive cancer; some lesions follow a biologically indolent course and do not cause symptoms or mortality [6]. Accordingly, there is an increasing need for more individualized and risk-based management strategies to prevent unnecessary treatment, especially for low-grade DCIS. Some incomplete studies such as the COMET, LORIS, and LORD trials are investigating the safety and benefit of active surveillance in patients with low-grade DCIS [5,7,8]. Therefore, it is important to differentiate low-grade DCIS from high-grade DCIS preoperatively, as it will prevent overtreatment. In addition, the most important treatment-related risk factor for recurrence in DCIS is the presence of residual DCIS [9]. Therefore, obtaining precise information about the size and distribution of DCIS in preoperative treatment planning is essential to prevent positive surgical margins and local regional recurrences.

Imaging plays a central role in this process. DCIS is detected as microcalcifications on mammography in 60–90% of cases [10,11]. In the literature, the sensitivity of mammography in detecting DCIS reaches 75% [12]. However, the sensitivity of mammography decreases, especially in dense breasts or in cases of DCIS without calcification [13].

Another effective factor in both DCIS detection and size assessment is the grade of DCIS [3]. There are also studies reporting that mammography underestimates tumor size [2]. Several studies have shown that magnetic resonance imaging (MRI) with dynamic contrast is the imaging modality with the highest sensitivity in the diagnosis of DCIS [10,13]. However, the MRI assessment of tumor size is still challenging, as DCIS is often seen as non-mass enhancement (NME) [10]. Therefore, there are still uncertainties regarding the use of breast MRI as a preoperative screening modality and the size estimation of DCIS.

Accordingly, the question of how well the imaging findings obtained by preoperative mammography and MRI are consistent with histopathologic facts is becoming increasingly important for both radiologists and breast surgeons on an international scale. Comprehensive analyses in this context are of global value in terms of individualizing treatment planning, avoiding unnecessarily aggressive approaches, and ensuring more efficient use of healthcare resources.

Based on this, the aim of our study was to evaluate the relationship between size and imaging features detected on preoperative mammography and MRI, and histopathological size and nuclear grade in patients with pure DCIS. The results obtained can guide clinical practice both nationally and internationally and aim to contribute to a more effective and individualized management strategy in breast cancer.

## 2. Materials and Methods

### 2.1. Patient Selection

This retrospective study was approved by the Institutional Review Board of our hospital (2023/142). Since the study was retrospective, informed consent by patients and providers was not required. Between 2015 and 2023, 210 patients who were operated on in our clinic and were diagnosed with DCIS were retrospectively analyzed. Patients with microinvasive components (68 patients), those with a history of breast cancer (4 patients), and patients who did not undergo mammography and breast MRI examinations or for whom the examinations provided insufficient diagnostic quality (48 patients) were excluded. A total of 90 patients with preoperative mammography and breast MRI examinations and histopathologically diagnosed pure DCIS were included in this study (Figure 1). Tumor size on mammography and MRI was measured and compared with histopathological tumor size as the gold standard.

### 2.2. Breast Imaging Technique

Standard craniocaudal and mediolateral oblique mammograms were obtained for all patients (Mammomat Revelation, Siemens, Erlangen, Germany). MRI was performed with a 1.5 Tesla MRI (Magnetom Aera, Siemens, Germany) using a 4-channel breast coil. Both breasts were examined with the patient in a prone position. Axial T1-weighted pre-contrast images (TR: 476 ms, TE: 11 ms, thickness: 4 mm, matrix 320 × 320, field of view 260 mm, flip angle 20°) and fat-suppressed T2-weighted images (TR:2250 ms, TE:56 ms, 4 mm thickness, matrix 320 × 320, field of view 300 mm, flip angle 20°) were obtained on all MRI scans. Axial diffusion-weighted images (DWI) were also obtained at b-values of 0, 500, and 1000 s/mm^2^ (TR/TE: 5600 ms/87 ms, field of view: 5 × 35 cm; matrix 220 × 102, slice thickness: 4 mm, slice gap: 0). Apparent diffusion coefficient (ADC) maps were automatically generated on the operating console by using the least squares method with all three images and b-values of 0, 500, and 1000 s/mm^2^. After intravenous injection of 0.1 mmol/kg body weight gadolinium contrast medium (Gadoteric acid, Dotarem, Guerbet), 5 dynamic postcontrast fat-suppressed T1-weighted images (TR: 4.53 ms, TE: 1.82 ms, 2 mm thickness, matrix 512 × 512, field of view 300 mm, flip angle 20°) were obtained. Subtracted images (contrast-enhanced minus unenhanced images) were obtained for all dynamic phases. Dynamic curves of percent enhancement versus time were obtained for lesions at small regions of interest, positioned on the brightest portion of the lesion. Multiplanar reconstructions and maximum intensity projections of subtracted images were obtained when necessary.

### 2.3. Data Analysis

Evaluation of the findings was performed in accordance with the Breast Imaging Reporting and Data System (BI-RADS) version 5. On mammography, breast density (category A = almost entirely fat, category B = scattered fibroglandular densities, category C = heterogeneously dense, and category D = extremely dense), presence, distribution (diffuse, regional, grouped, linear, and segmental), and type of microcalcifications (amorphous, coarse heterogenous, fine pleomorphic, and fine linear branching) were recorded according to BI-RADS.

On MRI, background parenchymal enhancement was categorized as minimal, mild, moderate, or marked according to BI-RADS. According to the fifth edition of the MRI BI-RADS descriptors, the morphology of the lesion was described as mass, NME, and focus. The distribution (focal, linear, segmental, regional, multiple regions, and diffuse) and internal enhancement patterns (homogeneous, heterogeneous, clumped, and clustered ring) of NME lesions were determined. The time signal curve type (persistent, plateau, and washout) of the lesion was recorded. Tumor size (largest diameter) was also recorded on both mammography and MRI. If the lesion was not visible on mammography and MRI, it was recorded as non-visible. All mammograms and MRIs were retrospectively reviewed in consensus by one radiologist with 30 years of experience and by one radiologist with 8 years of experience in breast imaging.

The postoperative histopathology results of the patients were reviewed by a breast pathologist with 20 years of experience. The DCIS cases were classified according to nuclear grade as low, intermediate, or high. Low- and intermediate-grade tumors were selected as the low-grade group, and high-grade tumors were selected as the high-grade group. The largest tumor diameter was set as the histological size of the tumor and reported in millimeters. In the presence of more than one focus, the largest focal size was set as the tumor size.

### 2.4. Statistical Analysis

Data analysis was performed with SPSS 26 and MedCalc19 versions. The suitability of continuous variables for normal distribution was investigated by means of graphical research, normality tests, and sample size. In conditions that did not provide normal distribution, comparisons of independent groups were performed with the Mann–Whitney U test. Median and interquartile range IQR (25–75%) values were given.

Categorical independent variables were presented as frequencies and percentages with cross tables. Distributions of independent groups were compared with the chi-square or Fisher’s exact test methods. Univariate odds ratios were calculated. Significant variables were included in the Multivariate Logistic Regression analysis as independent variables. Multivariate odds ratios were calculated with the Backward Stepwise (Wald) method. The compatibility of tumor sizes measured by mammography and MRI was examined with the concordance correlation coefficient method.

The difference between tumor sizes measured via mammography and MRI, according to the pathological tumor size, was calculated. The mean value of this variable was established using the one-sample t-test method according to the zero value. In all statistical comparison tests, the type 1 error margin was determined as α = 0.05 and was tested with two tails.

## 3. Results

All patients included in this study were female, and the mean age was 52.6 years (SD, ±9.8; range, 31–81 years). Of the 90 patients, 22 (24.5%) had low-grade, 30 (33.3%) had intermediate-grade, and 38 (42.2%) had high-grade DCIS. There was no statistically significant difference in nuclear grade according to age (*p* = 0.945). DCIS was detected in 82 (91.1%) of the 90 patients via mammography and 86 (95.5%) via MRI. Of the eight cases not detected using mammography, six were in the low-grade group and were detected using MRI. All four cases that were not detected using MRI were in the low-grade group and were seen as microcalcifications on mammography. No cases of DCIS failed to be detected on both mammography and MRI. Case examples are shown in Figure 2, Figure 3 and Figure 4. According to the BIRADS category, 73% of the patients in the 4A and 4B groups were in the low-grade group, while 57.9% of the patients in the BIRADS 4C and 5 categories were in the high-grade group (*p*: 0.006). BIRADS score was statistically significant in univariate and multivariate analyses (Table 1).

### 3.1. Mammographic Findings

The mammographic breast density categories were A in 4 patients, B in 44 patients, C in 34 patients, and D in 8 patients. Of the eight lesions that could not be seen on mammography, six had a category D breast density, but there was no statistically significant difference between the groups in terms of mammographic detectability when divided into two groups as A-B and C-D (*p* = 0.139). Of the 82 patients, 70 (85.4%) had microcalcifications, 40 (48.8%) had asymmetry, and 4 (4.9%) had architectural distortion. When all cases were analyzed, the most common type of microcalcification was amorphous (34/70). Fine pleomorphic and fine linear branching microcalcifications were observed in 42% of high-grade DCIS, while the most common microcalcification type in the low-grade group was amorphous (42%). There was a statistically significant difference between the groups in terms of the distribution of microcalcification types (*p* = 0.043). When the distribution of microcalcifications was analyzed, a grouped distribution was most common in low-grade DCIS cases (69%), while this rate was 37% in high-grade DCIS cases (*p* = 0.005). The localization of microcalcifications was statistically significant in univariate and multivariate analyses (OR: 3.86; OR: 4.64, respectively, Table 1).

### 3.2. Magnetic Resonance Imaging Findings

The most common background contrast enhancement on MRI was moderate, observed in 38 patients (42.2%), followed by mild, observed in 36 patients; minimal, observed in 14 patients; and marked, observed in 2 patients. MRI was able to detect DCIS in 86 patients (86/90, 95.5%). Background contrast enhancement was moderate and marked in the four patients not detected using MRI (*p* = 0.036). Mass enhancement was seen in 34.4% (31/90) and non-mass enhancement in 61.1% (55/90) of the lesions. There was no significant difference between the groups in terms of mass and non-mass contrast enhancement (*p* = 0.245). The most common distribution of NMEs was segmental in 60% of cases (33/55), and the most common contrast enhancement pattern was heterogeneous in 32.7% of cases (18/55). Diffusion restriction was seen in 36.7% (33/90) of the lesions. Diffusion restriction was statistically significantly higher in high-grade DCIS (*p* = 0.043). Time–signal intensity curves were persistent in 44.4% (40/90), plateau in 44.4% (40/90), and washout in 11.2% (10/90) of lesions. No significant correlation was found between background contrast enhancement, contrast enhancement distribution, internal enhancement patterns, time–signal intensity curve, and nuclear grade. The mammography and MRI findings are shown in Table 2.

### 3.3. Comparison of Imaging Methods and Pathological Size

The pathological median tumor size was 21 mm [interquartile range (IQR): 15–32], being 22 mm in the high-grade group and 20 mm in the low-grade group. There was no significant correlation between nuclear grade and tumor size (*p* = 0.082).

The median tumor size detected by means of mammography was 27 mm (IQR 17; 40). According to the pathological tumor size, the mean tumor size was 3.7 mm larger in the high-nuclear-grade group and 5 mm larger in the low-nuclear-grade group, which was not statistically significant (*p* = 0.60). The difference between mammographic measurements and pathological tumor size was not associated with breast density (*p* = 0.877) (Figure 5).

The median tumor size detected by means of MRI was 24 mm (IQR: 15; 36). The tumor size on MRI was 3.2 mm larger in the high-nuclear-grade group and 2.9 mm larger in the low-nuclear-grade group compared to the histopathological tumor size, and no significant difference was found between the groups (*p* = 0.217). There was no significant correlation between the difference between MRI measurements and pathological tumor size and background contrast (*p* = 0.120) (Figure 6).

According to Bland–Altman plot analysis, the mean difference in mammographic tumor size from pathological size was 4.43 mm [95% CI: 3.01–5.83] and the mean difference in MRI-measured size was 3.03 mm [95% CI: 2.12–3.95], both of which were statistically significantly higher (both *p* < 0.001) (Table 3, Figure 5). Although a cut-off value could not be calculated statistically, it was noticeable in the graphs that the difference in the pathological size increased as the tumor size increased.

The concordance correlation coefficient was 0.872 (95% CI: 0.8142–0.9126) between mammography and pathology, which was weak, and 0.939 (0.9101–0.9587) between MRI and pathology, which was moderate (Table 4, Figure 7).

## 4. Discussion

Our study examined the imaging findings of surgically resected lesions with pure DCIS and investigated the mammographic and MRI features that distinguish high-grade from low-grade lesions. We also compared the surgically proven histopathological size with the size measured by mammography and MRI and evaluated the concordance between them. MRI was found to be superior to mammography both in detecting DCIS and in estimating the actual size of the lesion.

In the literature, nuclear grade has been consistently associated with poor prognosis and local recurrence in DCIS [14]. In our study, a high BIRADS score; the presence of regional, linear, and segmental pleomorphic microcalcifications on mammography; and diffusion restriction on MRI were found to be associated with high-grade DCIS. Similarly to previous studies, our study showed that MRI was superior to mammography in the detection of DCIS, with a rate of 95.5% [15,16,17]. The sensitivity rates of mammography have been reported to be quite variable. Menell et al. [18] found the sensitivity of mammography to be 27%, while Schouten van der Velden et al. [19] found it to be 91%. In a study by Petrillo et al. comparing mammography and ultrasound with MRI performed in addition to these two methods, it was observed that the diagnostic sensitivity increased to 98.5%, with an increase of 10.1%, when MRI was added [20]. In addition, the cases that could not be detected by MRI were DCIS cases in the low-grade group in our study, similar to Prebisch et al. This suggests that MRI is more successful in the diagnosis of high-grade DCIS [9].

In a study by Jansen et al., 49 (75%) out of 65 DCIS cases had microcalcifications, of which 63.2% (31/49) were thin pleomorphic, fine linear, or fine linear branching calcifications [21]. According to the BI-RADS Atlas, fine linear branching is the most specific calcification morphology with a positive predictive value of 70% for malignancy on mammography. Fine pleomorphic microcalcifications are also high-risk. However, less suspicious morphologies such as amorphous and coarse heterogeneous can also be seen in DCIS [22]. In our study, the most common microcalcification morphology was amorphous, while pleomorphic microcalcifications were found to be associated with high-grade DCIS.

The study by Petrillo et al. showed that the rate of DCIS detection by MRI increased with nuclear grade [20]. Similarly, in our study, four cases of DCIS that could not be detected on MRI were found to be in the low-grade group. Jansen et al. found the predominant MRI features of pure DCIS lesions to be non-massive and clustered, with heterogeneous or homogeneous contrast enhancement conforming to segmental or linear distribution [21]. They stated that the reason for this was the accumulation of gadolinium in the ducts. In our study, NME was seen more frequently than mass contrast enhancement, with a rate of 61%. In DCIS, segmental, focal, and linear NME distribution is more common than regional and diffuse distribution. In addition, a clumped and clustered ring enhancement pattern is thought to be specific for DCIS [23]. In our study, segmental and regional distribution was more common, and focal and diffuse distribution was not observed. A heterogeneous and clustered enhancement pattern was more common than a homogeneous and clustered ring enhancement. Mass contrast enhancement and NME were not correlated with DCIS grade (*p* = 0.245). The kinetic curve pattern seen in DCIS is highly variable. The kinetic curve is related to the diffusion of contrast medium into blood vessels associated with vascularity and perfusion into the extracellular space, and generally peaks later in DCIS than in invasive cancers. It often results in an intermediate initial phase and/or a delayed persistent or plateau phase [23]. In the study by Jansen et al. [21], the most common kinetic curve pattern was type 3, representing washout, whereas in our study, persistence and plateau were found with equal frequency and more frequently than type 3.

Treatment management in DCIS depends on tumor size, which is another determinant of prognosis in addition to grade. Which imaging modalities provide the most accurate results and should be used preoperatively in terms of cost-effectiveness is still a matter of debate. The National Comprehensive Cancer Network (NCCN) guidelines for the preoperative evaluation of non-invasive ductal carcinoma recommend only bilateral mammography; optional breast MRI may be used, but ultrasonography is not recommended [24]. Numerous studies in the literature have found that MRI is more accurate than mammography in predicting histopathological size [15,25,26,27,28].

A 2022 meta-analysis showed that MRI accurately estimated tumor size within a predefined margin of error in the majority of cases, while overestimating tumor size in most of the remaining studies. Meta-analytic results confirm these findings with moderate certainty, with a mean difference of 3.85 mm between pathology and MRI measurements [13]. Baur et al. reported that lesion sizes measured by MRI showed a moderate correlation (Spearman’s correlation coefficient, *r* = 0.74) with histopathologically measured lesion sizes [29]. The overall sensitivity of MRI in accurately determining disease extent has been reported to reach almost 89% compared to 55% with mammography alone [29,30]. In one study, MRI could accurately estimate the pathological size of DCIS within 5 mm in 60% of cases, compared to 38% with mammography [31]. In our study, the difference between mammography and pathological size was higher than that of MRI. According to Bland–Altman plot analysis, the mean difference between mammography and pathological size was 4.43 mm, while the mean difference between MRI and pathological size was 3.03 mm. Concordance correlation coefficient showed a poor correlation between mammography and pathology, and a moderate correlation between MRI and pathology. In addition, in this study, it was thought that breast density may affect mammographic visibility and was compared in this respect. Consistent with the study of Kim et al. [32], no correlation was found between breast density and the mammographic–pathological size difference. However, in the study by Prebisch et al., mammography overestimated tumor size in category A breast structure, underestimated tumor size in category B, and underestimated tumor size in categories C and D [9]. Similarly, no significant differences were found in the tumor size difference when considering that background contrast in MRI may cause inaccurate size measurement. In the study by Daniel et al., mammography overestimated tumor size more than MRI. They explained this difference by the ability of mammography to detect thin, multifocal, and multicentric suspicious microcalcifications over larger areas. This study also showed that tumor size was correctly estimated by mammography in 52.4% of cases and overestimated in 41.4% of cases [28].

This study has some limitations. The first of these is the retrospective design of the study and the fact that the evaluation was based on available images. Mammographic compression standardization is unknown. There is insufficient information about the menstrual phase and hormone and drug use that may have affected background contrast enhancement on MRI. Secondly, the number of patients was relatively small. Thirdly, all patients included in the study were operated on and found to have pure DCIS. It could not be evaluated whether preoperative size and grade estimation would change the decision and type of surgery.

This study is one of the most recent and comprehensive studies in the literature on the prediction of both tumor size and nuclear grade with preoperative imaging in pure DCIS cases. Although several studies in the literature have reported that MRI has a higher sensitivity and more accurate tumor size prediction than mammography in the diagnosis of DCIS, our study demonstrates an innovative approach in terms of revealing the association of new parameters, such as diffusion restriction with high-grade DCIS. While other studies have generally focused on the sensitivity of MRI in the diagnosis of DCIS, our study analyzed the relationship of both morphological and kinetic features with nuclear grade in detail and supported it with advanced statistical methods. In all these aspects, we believe that our study is innovative enough to contribute to clinical decision support systems and fills an important gap in the literature.

## 5. Conclusions

Our study showed that MRI was more accurate than mammography in detecting the presence and size of DCIS. Fine pleomorphic and fine linear branching microcalcifications with segmental, linear, and regional distribution on mammography were associated with high-grade DCIS. We did not find any significant MRI features suggestive of high-grade DCIS, but diffusion restriction is an important finding to be considered. Prospective studies in larger populations are needed to determine how preoperative results will affect the diagnostic and therapeutic processes.

## Figures and Tables

**Figure 1 diagnostics-15-01801-f001:**
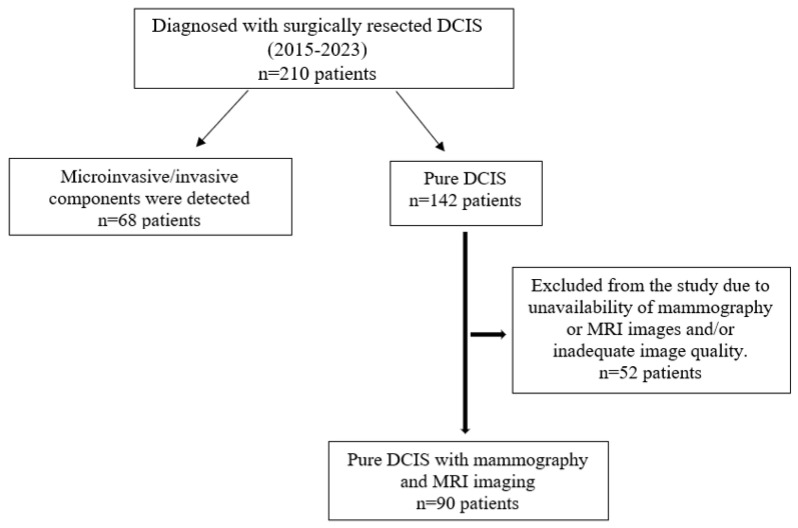
Flowchart of participants throughout this study.

**Figure 2 diagnostics-15-01801-f002:**
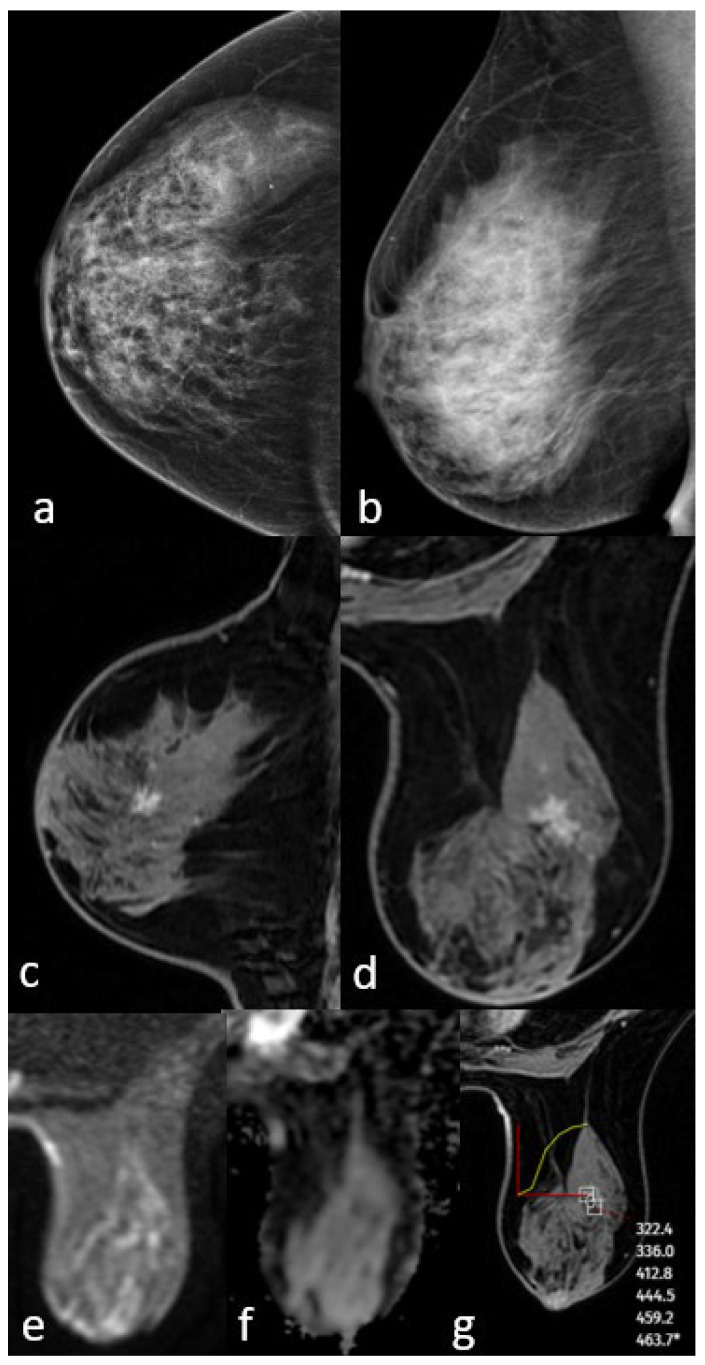
Mammography shows extremely dense breast density and no mass (**a**,**b**). MRI post-contrast T1-weighted images show an irregularly shaped mass with a spiculated contour (**c**,**d**). No significant diffusion restriction on diffusion-weighted imaging (**e**,**f**) and a persistent kinetic curve on the dynamic series (**g**). The pathology result is low-grade ductal carcinoma in situ.

**Figure 3 diagnostics-15-01801-f003:**
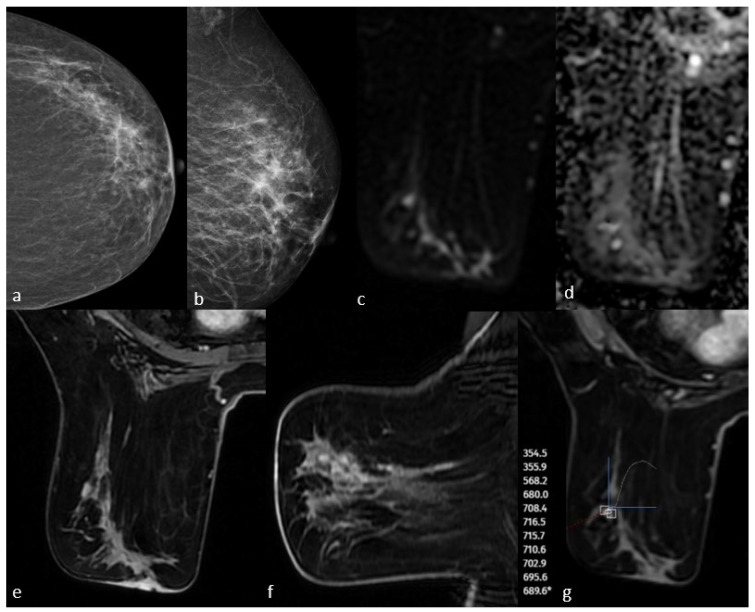
Mammography shows fine linear branching microcalcifications in the upper outer quadrant (**a**,**b**). Post-contrast T1-weighted images on MRI show non-massive contrast enhancement corresponding to this area (**e**,**f**). Diffusion-weighted imaging shows no significant diffusion restriction (**c**,**d**), and dynamic series shows a plateau kinetic curve (**g**). The pathology result is high-grade ductal carcinoma in situ.

**Figure 4 diagnostics-15-01801-f004:**
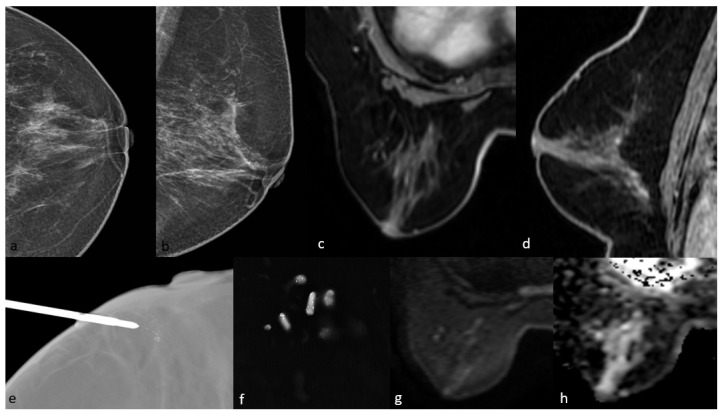
Mammography shows grouped amorphous microcalcifications in the periareolar area (**a**,**b**). On MRI, there is no contrast enhancement in this area on post-contrast T1-weighted images (**c**,**d**) and no significant diffusion restriction on diffusion-weighted imaging (**g**,**h**). The pathology result of mammography-guided biopsy (**e**,**f**) of the microcalcifications was low-grade ductal carcinoma in situ.

**Figure 5 diagnostics-15-01801-f005:**
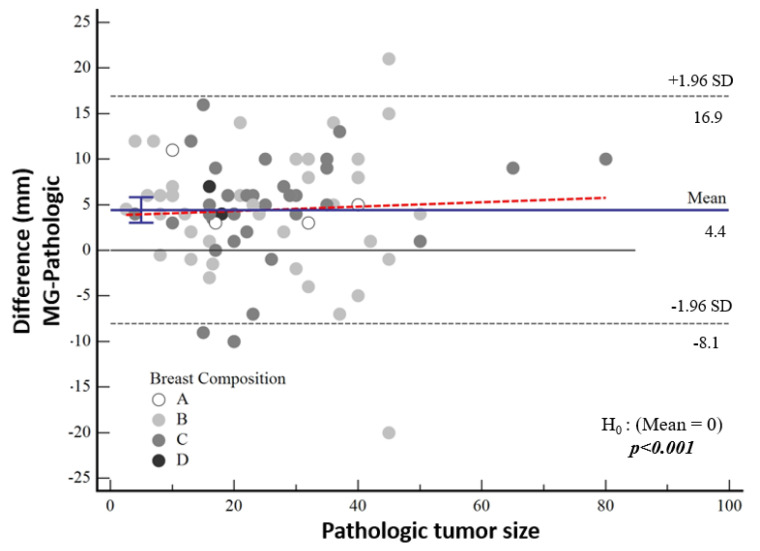
Bland–Altman plot demonstrating the distribution of the difference between mammographic and pathological sizes according to breast composition.

**Figure 6 diagnostics-15-01801-f006:**
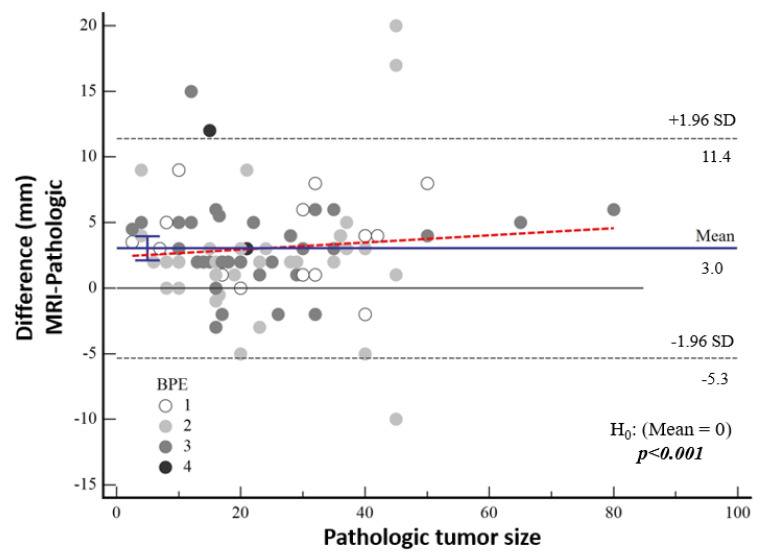
Bland–Altman plot showing the distribution of the difference between the magnetic resonance imaging and pathological sizes according to background contrast enhancement.

**Figure 7 diagnostics-15-01801-f007:**
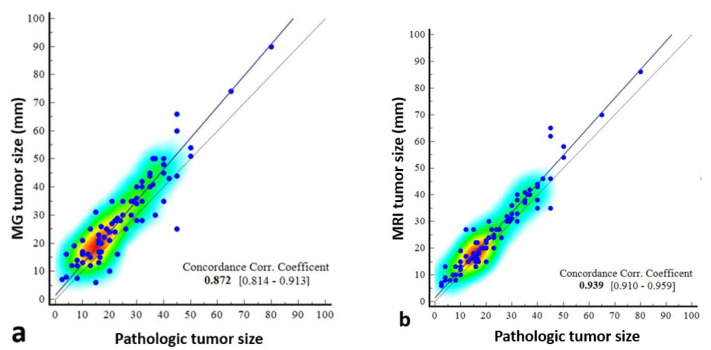
Scatter dot plots showing the correlation of pathologically measured tumor size with mammography- and magnetic resonance imaging (MRI)-measured size. The correlation coefficient of concordance is poor between mammography and pathology (**a**), and the scatterplots indicate that there may be deviations, especially in certain size ranges. Between MRI and pathology (**b**), it is moderate, indicating that MRI provides a reliable estimate of tumor size, especially in the 0–80 mm range.

**Table 1 diagnostics-15-01801-t001:** Risk analysis for high-grade DCIS.

Variable	Reference	Univariate		Multivariate	
		OR [95%CI]	*p* ^a^	OR [95%CI]	*p* ^b^
Microcalcification distribution	(Regional-Linear-Segmental/Grouped-None)	3.86 [1.59–9.34]	**0.003**	4.64 [1.76–12.21]	**0.002**
BI-RADS	(3-4A-4B/4C-5)	3.73 [1.53–9.08]	**0.004**	4.52 [1.70–12.00]	**0.003**
Diffusion Restriction	(Yes/No)	2.71 [1.12–6.56]	**0.027**	—	ns
Breast composition	(A-B/C-D)	2.20 [0.94–5.16]	0.070	—	ns

**^a^** Mantel–Haenszel Common Odds Ratio Estimate; **^b^** Logistic Regression Method = Backward Stepwise (Wald); OR = Odds Ratio, ns = Not Significant; Constant: −1.66.

**Table 2 diagnostics-15-01801-t002:** Mammography and magnetic resonance imaging findings according to nuclear grade.

Variable		Total		High-Grade Group		Low-Grade Group		*p*
		n	%	n	%	n	%	
BI-RADS category	3-4A-4B 4C-5	54 36	60% 40%	16 22	42% 58%	38 14	27% 73%	**0.006**
Breast composition	Type A-B Type C-D	48 42	53% 47%	16 22	42% 58%	32 20	62% 38%	0.107
Microcalcification morphology	Amorphous Pleomorphic Uniform None	34 24 12 20	38% 27% 13% 22%	12 16 4 6	32% 42% 11% 16%	22 8 8 14	42% 15% 15% 27%	**0.043**
Microcalcification distribution	Regional-Linear-Segmental Grouped-None	40 50	44% 56%	24 14	63% 37%	16 36	31% 69%	**0.005**
Architectural distortion	Yes No	4 86	4% 96%	2 36	5% 95%	2 50	4% 96%	1.000
Background parenchymal enhancement	1 2 3-4	14 36 40	16% 40% 44%	4 14 20	11% 37% 53%	10 22 10	19% 42% 38%	0.329
MRI morphology	Mass Non-Mass Enhancement	31 59	34% 66%	10 28	26% 74%	21 31	40% 60%	0.245
Non-mass enhancement distribution	Linear Segmental Regional	4 33 18	7% 60% 33%	4 19 9	13% 59% 28%	0 14 9	0% 61% 39%	na
Diffusion restriction	Yes No	33 57	37% 63%	19 19	50% 50%	14 38	27% 73%	**0.043**
Kinetic curve type	Persistent Plateau Washout	40 40 6	47% 47% 6%	13 21 4	34% 55% 11%	27 19 2	56% 40% 4%	na

BI-RADS: Breast Imaging Reporting and Data System; MRI: magnetic resonance imaging.

**Table 3 diagnostics-15-01801-t003:** Mean of pathological tumor size difference.

	Diff. Mean (mm) (95%CI)	*p* (H0: Mean = 0)
Mammographic tumor size (mm)	4.43 [3.01–5.83]	<0.001
MRI tumor size (mm)	3.03 [2.12–3.95]	<0.001

**Table 4 diagnostics-15-01801-t004:** Concordance correlation coefficient.

	Pathologic Tumor Size (mm)	
Mammographic tumor size (mm)	0.872 (0.8142–0.9126)	Poor
MRI tumor size (mm)	0.939 (0.9101–0.9587)	Moderate

Value of concordance correlation coefficient: <0.90, poor; 0.90–0.95, moderate; 0.95–0.99, substantial; and >0.99, almost perfect.

## Data Availability

The datasets used and/or analyzed during the current study are available and can be obtained from the corresponding author upon reasonable request.

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
