# Peer review of "Can Mammography and Magnetic Resonance Imaging Predict the Preoperative Size and Nuclear Grade of Pure Ductal Carcinoma In Situ?"

_diagnostics, 2025, doi:10.3390/diagnostics15141801_

Round 1
Reviewer 1 Report
Comments and Suggestions for Authors
The core content of this article is the preoperative radiological assessment of ductal carcinoma in situ (DCIS), particularly exploring the accuracy of mammography and magnetic resonance imaging (MRI) in predicting the size and nuclear grade of pure DCIS. The study results are of significant clinical guidance for preoperative assessment of DCIS and can help optimize treatment decisions. Although MRI has higher sensitivity in detecting lesions, its specificity is relatively low, which may lead to misdiagnosis. The study did not fully explore how radiological assessment affects preoperative treatment decisions, such as whether to change the surgical approach or treatment plan. Increasing the sample size can enhance the statistical power and reliability of the study results. MRI is more accurate than mammography in detecting the presence and size of DCIS. Fine pleomorphic and fine linear branching microcalcifications on mammography are associated with high-grade DCIS. Diffusion restriction on MRI is an important feature of high-grade DCIS. It is recommended to conduct prospective studies to better evaluate the impact of preoperative radiological assessment on treatment decisions. It is also suggested to further improve some of the language expressions for better clarity.
Author Response
Comments 1: The core content of this article is the preoperative radiological assessment of ductal carcinoma in situ (DCIS), particularly exploring the accuracy of mammography and magnetic resonance imaging (MRI) in predicting the size and nuclear grade of pure DCIS. The study results are of significant clinical guidance for preoperative assessment of DCIS and can help optimize treatment decisions. Although MRI has higher sensitivity in detecting lesions, its specificity is relatively low, which may lead to misdiagnosis. The study did not fully explore how radiological assessment affects preoperative treatment decisions, such as whether to change the surgical approach or treatment plan. Increasing the sample size can enhance the statistical power and reliability of the study results. MRI is more accurate than mammography in detecting the presence and size of DCIS. Fine pleomorphic and fine linear branching microcalcifications on mammography are associated with high-grade DCIS. Diffusion restriction on MRI is an important feature of high-grade DCIS. It is recommended to conduct prospective studies to better evaluate the impact of preoperative radiological assessment on treatment decisions. It is also suggested to further improve some of the language expressions for better clarity.
Response 1: Thank you for your valuable comment. The important points you have made are detailed in the findings and discussion. In addition, the relatively small number of patients and the retrospective design of the study are also mentioned in the article as our limitations. According to your suggestion, the English language has been edited for better comprehensibility.
Reviewer 2 Report
Comments and Suggestions for Authors
In the manuscript entitled “Can mammography and magnetic resonance imaging predict preoperative size and nuclear grade of pure ductal carcinoma in situ?” the authors present several important results. Although the results are well presented, there are some suggestions that should be addressed before I can recommend the manuscript for publication in Diagnostics (MDPI journal):
- I recommend enhancing the Introduction section to provide the reader with a broader context regarding the current use of mammography and magnetic resonance imaging in predicting preoperative data. It is important to clearly emphasize the international relevance of this topic.
- The methodology section does not clearly explain why a sample size of 90 was selected. Is this sample size sufficiently representative to support the conclusions?
- Figures should be described in greater detail, as some lack sufficient context and may cause the reader to miss important information—for example, Figure 7.
- The figures also require careful revision. For instance, the x-axis label in Figure 7 is broken and difficult to read.
- The text in Table 2 is overlapping and illegible.
- The text on the axes of several figures is too small and cannot be easily read.
- A section should be added before the Conclusions to highlight the novelty of this work in comparison with previously published studies.
Author Response
Comments 1: I recommend enhancing the Introduction section to provide the reader with a broader context regarding the current use of mammography and magnetic resonance imaging in predicting preoperative data. It is important to clearly emphasize the international relevance of this topic.
Response 1: Thank you for your valuable comment. In line with your suggestion, the Introduction has been expanded and elaborated. Accordingly, sequential changes have been made in the references.
Comments 2: The methodology section does not clearly explain why a sample size of 90 was selected. Is this sample size sufficiently representative to support the conclusions?
Response 2: Thank you for your comment. Because our study was designed to be retrospective and to investigate patients between certain years (2015-2023), the number of patients was 90. Patients before 2015 were not included in the study because the MRI quality in our clinic was not suitable for evaluation. Of course, a larger number of patients would make our study statistically stronger. We mentioned this as a limitation.
Comments 3: Figures should be described in greater detail, as some lack sufficient context and may cause the reader to miss important information—for example, Figure 7.
Response 3: Thank you for your attention. Detailed explanations about the figures have been added to the article.
Comments 4: The figures also require careful revision. For instance, the x-axis label in Figure 7 is broken and difficult to read.
Response 4: Thank you for your suggestion. The figures have been revised, and the text with broken labels and hard-to-read text has been reorganized.
Comments 5: The text in Table 2 is overlapping and illegible.
Response 5: Thank you for your comment. The location of Table 2 has been edited to prevent overlapping.
Comments 6: The text on the axes of several figures is too small and cannot be easily read.
Response 6: Thank you for pointing this out. We agree with this comment. The text in the figure has been corrected.
Comments 7: A section should be added before the Conclusions to highlight the novelty of this work in comparison with previously published studies.
Response 7: We agree with you sincerely. This was an important point that we overlooked; thank you for your contribution. A relevant paragraph has been added to the article before the conclusion.
Round 2
Reviewer 1 Report
Comments and Suggestions for Authors
I have no questions.
Reviewer 2 Report
Comments and Suggestions for Authors
The authors have addressed the comments and suggestions. Therefore, I recommend accepting the manuscript for publication.